# Mechanochemical Preparation of a Novel Slow-Release Fertilizer Based on K₂SO₄-kaolinite

**Ehab AlShamaileh** [1,*], **Mohammad Alrbaihat** [2], **Iessa Moosa** [1], **Qusay Abu-Afifeh** [1,3], **Hebah Al-Fayyad** [1], **Imad Hamadneh** [1] **and Aiman Al-Rawajfeh** [4]

1   Department of Chemistry, The University of Jordan, Amman 11942, Jordan
2   Emirates School Establishment, Dubai 3962, United Arab Emirates
3   Department of Land, Water and Environment, The University of Jordan, Amman 11942, Jordan
4   Department of Natural Resources and Chemical Engineering, Tafila Technical University, Tafila 66110, Jordan
*   Correspondence: ehab@ju.edu.jo

**Abstract:** In this study, a novel slow-release fertilizer (SRF) consisting of kaolinite and K₂SO₄ was prepared, employing the process of mechanochemical milling in a planetary ball mill. To obtain the optimum milling time and speed, several samples were made at fixed mass ratios of kaolinite: K₂SO₄ (3:1). The milling rotational speed ranged from 200 to 700 rpm for 120 min. Different milling times ranging from 60 to 180 min at fixed 600 rpm milling speed were also investigated to evaluate the incorporation of K₂SO₄ and to measure the liberation of $K^+$ and $SO_4^{2-}$ ions into solution. The properties of the studied samples were analyzed by Fourier transformation infrared spectrometry (FTIR), thermal gravimetric analysis (TGA), and ion chromatography (IC). The mechanochemical process is a green chemistry procedure that is successfully applied to incorporate K₂SO₄ into the amorphous kaolinite structure. The slow-release performance was evaluated by determining the $K^+$ and $SO_4^{2-}$ content in the aqueous solution upon leaching. The optimum released amount of $K^+$ after 24 h was 32 mg L$^{-1}$ for the milling conditions of 180 min and 700 rpm, indicating that K₂SO₄-kaolinite has good slow-release properties. The novel SRF is cost-effective, environmentally friendly, and improves the fertilizer's efficiency in many agricultural applications.

**Keywords:** kaolinite; mechanochemical milling; crops; soil; plants; green chemistry; fertilizer efficiency





## 1. Introduction

The world population is continuously and rapidly growing, resulting in higher demands for agricultural produce. Due to this humanitarian issue, modern agriculture plays a fundamental role in satisfying the need for food and other agricultural products [1]. To get better agricultural yields, soil fertility is usually improved by the input of large quantities of fertilizers [2]. Fertilizers and water are the most important substances in agricultural production. Nitrogen (N), phosphorus (P), and potassium (K) are three major chemical elements required for the growth of plants. Currently, commercial fertilizers feature instant dissolution [3–6]. However, since the rate of release of nutrients is much higher than the rate of adsorption from crops, traditional chemical fertilizers often cause many problems, such as the waste of resources and environmental pollution. Moreover, long-term use of these fertilizers can lead to soil compaction and reduced crop yields [7,8].

According to Mebdoua, a massive quantity of fertilizers was used in farms worldwide during the past few decades [9]. Although potassium (K) is a vital element for plant growth, it has not been given enough attention compared with nitrogen and phosphorous in many crop production systems. Potassium plays an important role in the activation of enzymes, internal cation-cation balance, water absorption, osmoregulation, translocation of photosynthate, and protein synthesis, which are essential in plants' growth and health alike. The element is critical in the production of conventional and organic crops. Generally,

plants' stress, such as from frost, drought, heat, and increasing light absorption is improved with appropriate K nutrition [10].

There are several K chemical sources such as potassium sulfate ($K_2SO_4$) and potassium chloride (KCl). Both chemicals are known to be good sources of K as well as $SO_4^{2-}$ and $Cl^-$ that, in turn, provide essential nutrients for plants. $Cl^-$ is not recommended in large amounts due to its salinity. However, it has a key role in the health and resistance of plant diseases. In fact, Li et al. [11] stated that soil microbes are affected by the existence of $Cl^-$ and $SO_4^{2-}$ ions in the soil. It has been observed by Okur et al. that adding K element can decrease the effect of harmful soil salinity on the activity of microbes [12].

Slow-release fertilizers (SRFs) represent a promising approach to alleviate environmental pollution and improve nutrient utilization efficiency as well. The SRFs are produced to gradually release nutrient materials at the rate required by the plants, which is a satisfactory way to reduce the nutrient waste. The advantages of using SRFs instead of conventional fertilizers include higher efficiency of the fertilizer use, the continuous supply of nutrients for a prolonged period, and lower nutrient losses due to volatilization and leaching [1,13].

SRFs are usually prepared by either physical or chemical methods such as the dispersion of ordinary fertilizer in some matrix, or by the encapsulating of familiar fertilizer within the structure of some carriers. The state of diffusion-delayed nutrient release is extensively debated by many workers in the field in nutrient research [14,15].

Recently, AlShamaileh and other researchers have illustrated that a mechanochemical approach can be employed to produce many slow-release fertilizers (SRFs) [3,16]. Clay minerals such as kaolinite play an important role in many applications [17]. Al-Rawajfeh et al. have prepared kaolinite composites in conjugation with two compounds $KH_2PO_4$ and $(NH_4)_2HPO_4$ by a grinding process using a planetary ball milling machine [18]. They confirmed that mechanochemistry is an application process to prepare amorphous kaolinite, which performs as a $PO_4^{3-}$, $K^+$, and $NH_4^+$ ion carrier of nutrients to make a slow-release fertilizer. AlShamaileh et al. have reported the application of the mechanochemical route to produce some complex compounds to be used as a slow-release fertilizer [19]. Rudmin et al. have also investigated the production of complex materials starting from glauconite by a mechanochemical method to serve as slow-release fertilizers [15]. Mechanochemical reactions are those induced by mechanical means (milling, grinding, compression) and conducted either in solvent-free conditions or using catalytic amounts of solvent [20,21].

Cavallaro et al. confirmed that halloysite clays are natural nanoparticles formed by rolling kaolin sheets. Due to its versatile properties, such as hollow tubular morphology, tunable surface chemistry, and biocompatibility, kaolin was recently studied for development of innovative nanomaterials for biotechnological applications [22].

Makó et al. have studied the production of kaolinite–urea intercalation compounds by mechanochemical intercalation and found them to have crystallite sizes lower than those obtained by the aqueous solution method [23].

The current study aims to focus on the effect of milling conditions and parameters—particularly time and speed of milling—on the (kaolinite-$K_2SO_4$) system properties, attempting thereby to completely transform the kaolinite clay mineral into an amorphous phase and creating chemical bonding between K–Al–Si–P–O as an amorphous glass phase. Eventually, a water-insoluble SRF fertilizer will be gained with slow $K^+$ and $SO_4^{2-}$ ions release.

## 2. Materials and Methods

### 2.1. Materials and Mechanochemical Process

White kaolinite ($Al_2Si_2O_5(OH)_4$) (>90 mass%) was obtained from Wadi Araba, Jordan [24] and used as received. Potassium sulphate ($K_2SO_4$) was purchased from Merck, USA. The chosen materials were milled using a dry planetary ball milling (Pulverisette 7, Fritsch, Germany) with different rotational speeds and milling time. The planetary ball mill consists of two zirconia mill pots with an inner volume of 45 $cm^3$. Six balls made of zirconia with 15 mm diameter were used in the grinding process. All studied samples were prepared by co-grinding kaolinite with $K_2SO_4$ at different experimental conditions.

Two groups of experiments were done with 3:1 a mass ratio of kaolinite: $K_2SO_4$. This ratio was chosen because it reduces the amount of $K_2SO_4$ used as a fertilizer (100%) keeping a significant K concentration (~33%). Two experiments of milling were run as follows:

1. In the first one, the time of milling was kept constant with various milling speeds.
2. In the second one, the milling speed was fixed with different milling times.

The details of the above experiments are recorded in Table 1.

**Table 1.** Mixture ratios and conditions used in the experiments.

| Series | Sample (kaolinite-$K_2SO_4$) Weight Ratio | Milling Speed rpm | Milling Time min |
|---|---|---|---|
| Exp. 1 | 3:1 | 700 | 60, 120, 180 |
| Exp. 2 | 3:1 | 200, 400, 700 | 120 |

*2.2. Characterization*

The prepared samples were tested using the Fourier infrared spectrometer type (FTIR, NEXUS, EPS-870) in the wavenumber of range 500–4000 cm$^{-1}$ with a resolution of 2 cm$^{-1}$. In addition, the system of (TG-DTA, STA-409 PC, NETZSCH) was employed for differential thermal analysis in an $N_2$ gas atmosphere with a heating rate of 10 °C/min from room temperature to 1000 °C. X-ray diffraction patterns were obtained using a Philips PW-1710 automated diffractometer at 40 kV and 30 mA with a Cu tube operating at 5° to 60° Bragg angles, counting for 2 s between each angle. The release behavior of nutrients in solution from the prepared K–Si–Ca–O system was analyzed. In addition, leaching experiments on the ground samples in distilled water were conducted at room temperature in a 100-mL glass beaker, with 1 g of milled sample and 20 mL of distilled water.

The leaching time was set for 24 h. After this, the solid-liquid separation was carried out using a suction filtration facility of 0.45 µm pore size. The concentrations of $K^+$ and $SO_4^{2-}$ ions (nutrients) ions in the gained filtrated solution were measured by liquid ion chromatography (IC, Thermo scientific, column series Dionex, CS-5000+DP, Dreieich, Germany).

**3. Results and Discussion**

Potassium sulfate is considered as a good soluble fertilizer and an important source for $K^+$ and $SO_4^{2-}$ ions in the soil. Since nutrients are released at a slower rate throughout the season, slow-release fertilizers are excellent alternatives to the soluble fertilizers since plants can absorb most of the nutrients without much loss by leaching. This section focuses on the efficiency of different parameters in releasing $K^+$ and $SO_4^{2-}$ from prepared synthesized kaolinite-$K_2SO_4$ SRFs.

*3.1. Milling Time Effect on Mechanochemical Synthesis of kaolinite-$K_2SO_4$ SRFs*

The effect of milling time on the characteristics of the kaolinite-$K_2SO_4$ (3:1 mass ratio) from 60 to 180 min at a fixed milling speed was investigated. From the FTIR spectra illustrated in Figure 1, it can be seen that as the time of milling increases, the band intensity varies especially at the peak around 915 and 1115 cm$^{-1}$. The peak intensity gradually decreases going from a milling time of 60 min until it finally disappears at 180 min. The distinctive band of the Si-O, which stretches to approximately 1049 cm$^{-1}$ for high milling speed samples, becomes broader and shifts for milled samples to a higher frequency, as evidenced by the development of amorphous silica during the mechanochemical reaction.

Moreover, the peaks of the stretching vibrations of hydroxyl groups of kaolinite in the range 3619–3688 cm$^{-1}$ did not appear in the milled samples, which means that the intercalation takes place at the milling time of 60, 120, and 180 min with no distinguishable features in this region.

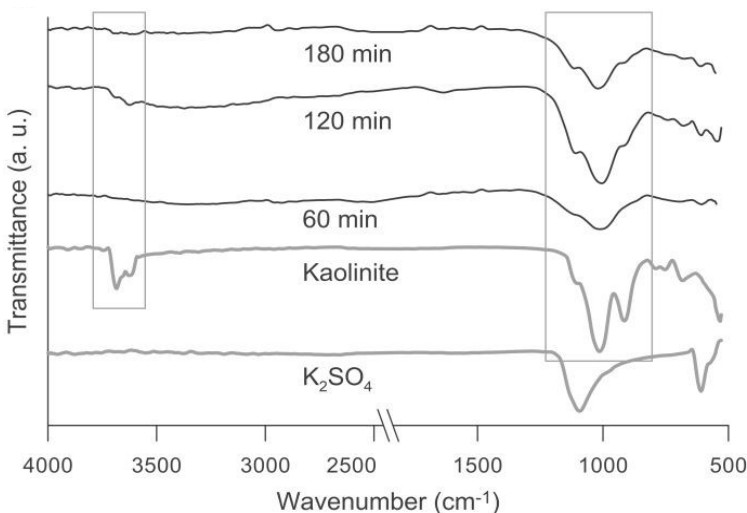

**Figure 1.** FTIR spectra of kaolinite-$K_2SO_4$ samples milled at 700 rpm with different milling times.

The XRD patterns in Figure 2 show that featured characteristic patterns corresponding to $K_2SO_4$ and kaolinite had disappeared in the milled products; this is particularly noticeable for the patterns at around 12.5 and 25 from the kaolinite pattern, which could suggest that $K_2SO_4$ has been incorporated into the amorphous structure of kaolinite [25–27]. The FT-IR and XRD results prove that the milling time is an important parameter for the mechanochemical synthesis process. The longer milling time makes the compound's destruction faster during grinding and allows the formation of amorphous phases of the mixed ingredients [15,16,26].

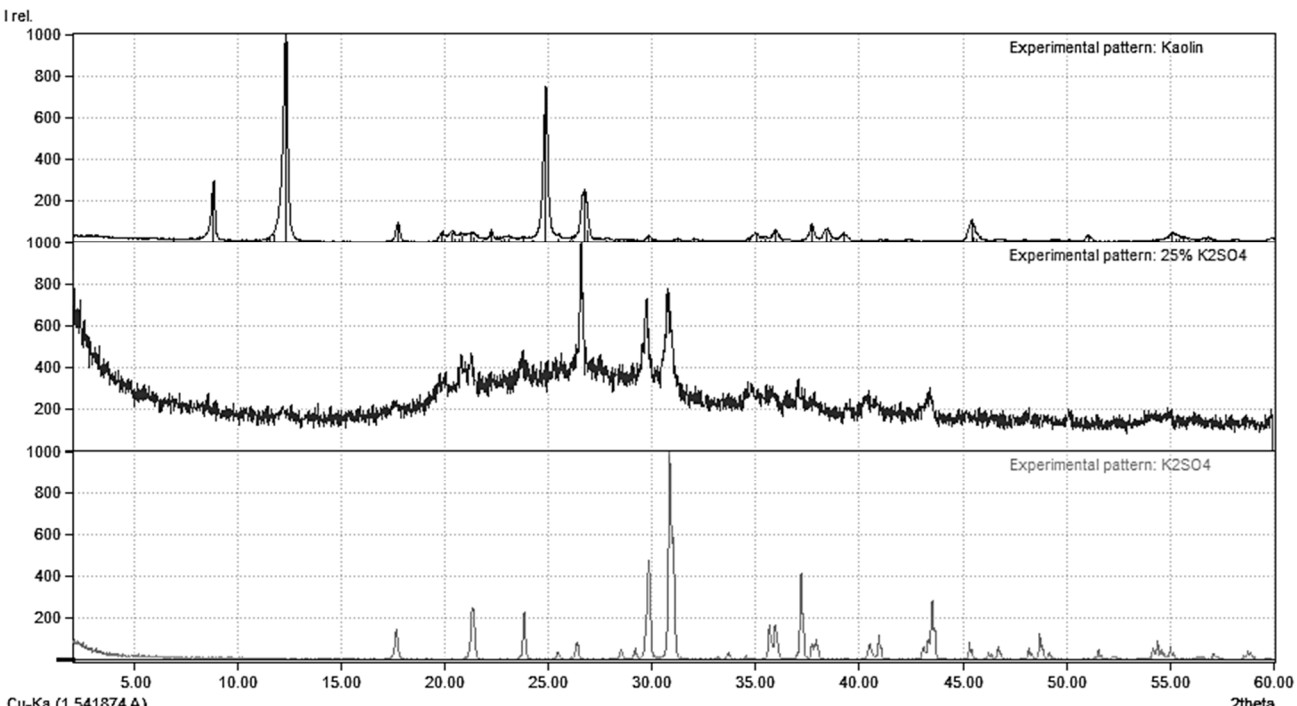

**Figure 2.** XRD patterns of kaolinite-$K_2SO_4$ system milled at speed 700 rpm for 120 min.

The curves of the thermogravimetric system revealed that the kaolinite-$K_2SO_4$ system decomposed in three steps, and showed decomposition (weight % change) peaks at temperatures of 100, 490, and 707 °C when milled at 700 rpm for three different times (60, 120, 180 min). Figure 3 shows a clear decomposition for the sample milled at 700 rpm for 180 min, from which it can be concluded that the intercalation reaction takes place better

at higher milling times. The TGA patterns also reveal the presence of only two mass loss events for the sample milled for 180 min. On the other hand, three mass loss events are seen for samples milled for 60, 120 min.

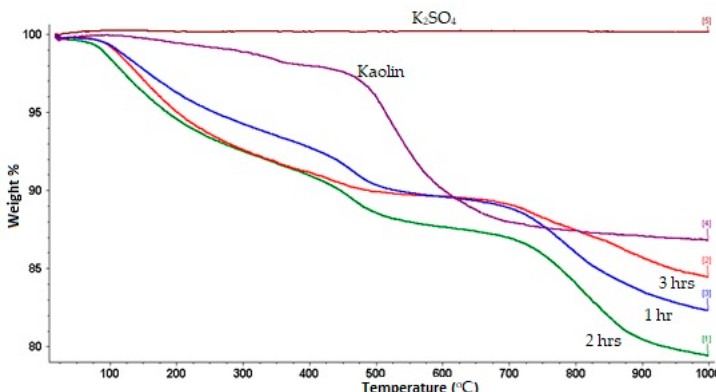

**Figure 3.** TGA patterns of kaolinite-$K_2SO_4$ (3:1 mass ratio), various milling times, 700 rpm.

Significant and clear effects are obtained for time milling for release of $K^+$ and $SO_4^{2-}$ nutrients from kaolinite- $K_2SO_4$ mixtures milled at 700 rpm, as shown in Table 2. The release rate of the nutrients decreases as the kaolinite content increases. The milling rotational speed was kept constant at 700 rpm for varied milling times during sample preparation.

**Table 2.** The concentration of $K^+$ and $SO_4^{2-}$ nutrients released from kaolinite-$K_2SO_4$ sample mixture after milling at 700 rpm for different milling times.

| Milling Time (min) | Conc. of $K^+$ (mg $L^{-1}$) | Conc. of $SO_4^{2-}$ (mg $L^{-1}$) |
| --- | --- | --- |
| 60 | 289.6 | 2801.0 |
| 120 | 36.0 | 2478.0 |

Figure 4 illustrates the release performance of the kaolinite-$K_2SO_4$ nutrient mixtures milled at 700 rpm as a function of milling time and dispersed in distilled water for 24 h. The released ions of $K^+$ and $SO_4^{2-}$ for the sample prepared at 60 min reached over 289.6 and 2801 mg $L^{-1}$, respectively. This indicates that the milling time of 60 min was enough to complete the incorporation of the $K_2SO_4$ compound into the amorphous structure of the used kaolinite. It is noticed that the release of both $K^+$ and $SO_4^{2-}$ nutrients sharply decreased into solution when the milling time increased from 60 to 120 min, particularly the $K^+$ ions which reached about 36 mg $L^{-1}$. Another significant decrease of around 32 and 2749 mg $L^{-1}$ of $K^+$ and $SO_4^{2-}$ ions from samples that produced with milling time of 180 min.

### 3.2. Rotating Milling Speed Effect on the Mechanochemical Synthesis of the kaolinite-$K_2SO_4$ SRFs

In this part of the experiment, the milling time of 120 min was fixed, whereas the rotational milling speeds were changed, within the range of 200 to 700 rpm. The effect of milling rotational speed on the amorphization of the kaolinite-$K_2SO_4$ (3:1 mass ratio) sample mixture was studied using FTIR (Figure 5). The FTIR spectra showed that the characteristic of $K_2SO_4$ spectra persisted in the milled samples with lower milling speed, namely in the range 200–400 rpm. Generally, at milling speed of 700 rpm, the starting compounds of kaolinite and $K_2SO_4$ were merged to form a unified amorphous structure.

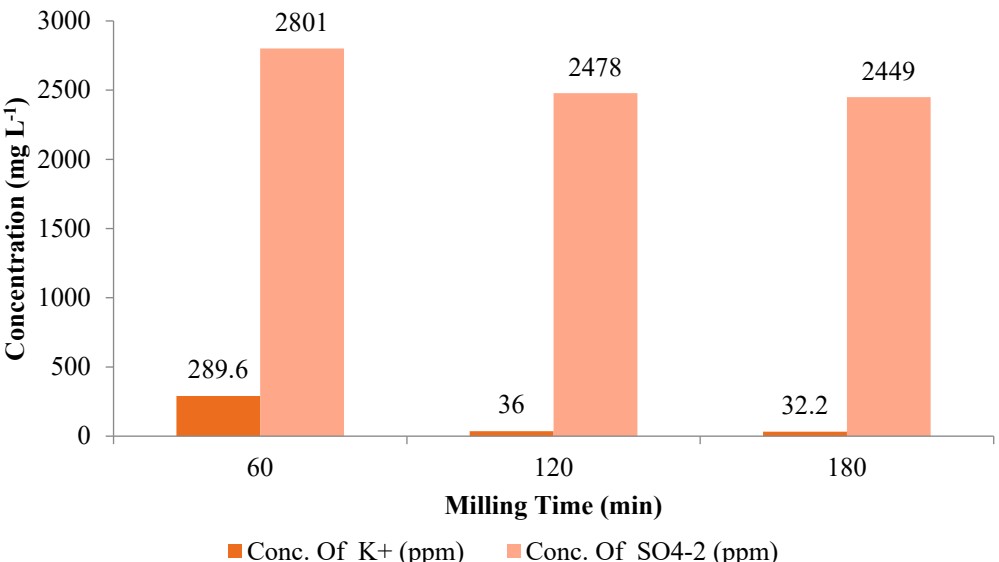

**Figure 4.** Profile of produced nutrient mixtures dispersed in distilled water at different milling times with constant milling speed 700 rpm.

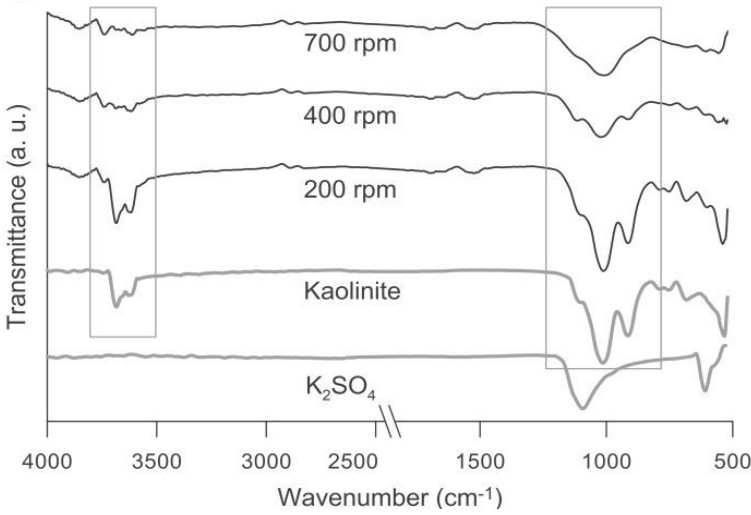

**Figure 5.** FTIR spectra of kaolinite-$K_2SO_4$ samples milled at different rotating speeds for 120 min.

The 1101 cm$^{-1}$ band of the $K_2SO_4$ spectrum is shifted up the field while increasing the milling speed for the kaolinite-$K_2SO_4$ samples until it reaches a minimum value in the 700-rpm sample, which indicates complete intercalation of $K_2SO_4$ in kaolinite. Furthermore, in the region between 3620–3700 cm$^{-1}$, two broad bands that belong to the OH stretching vibrations appeared in the kaolinite spectrum and in the kaolinite-$K_2SO_4$ samples that were milled at less than 400 rpm. We notice that some peaks have disappeared for the samples milled at over 400 rpm due to dehydroxylation. Our conclusion is that the mechanochemical predictable reaction takes place at higher speeds of milling rotation. However, it has been stated that the rate of milling rotation is considered the major factor in the nutrients' releasing process [19]. The obtained results in the current article confirm that the optimum rotational milling speeds are in the range of 400–700 rpm to produce $K_2SO_4$ fertilizers by employing the mechanochemical route.

Figure 6 shows that our measurement of the mass loss of about 12.3% is because of the kaolinite thermal dehydroxylation, which is a good value as it is near the predicted percentage of 13.96. This result confirms well the purity of the used kaolinite. As mentioned earlier, the milled $K_2SO_4$ decomposes basically in three steps by endothermic peaks at

temperatures of 100, 490, and 707 °C with a milling speed of 400 rpm. Furthermore, from Figure 5 it can be seen that the reaction of intercalation can take place better in two steps at the temperatures 100 °C and 700 °C, where the mixture is milled at a rotational speed of 700 rpm.

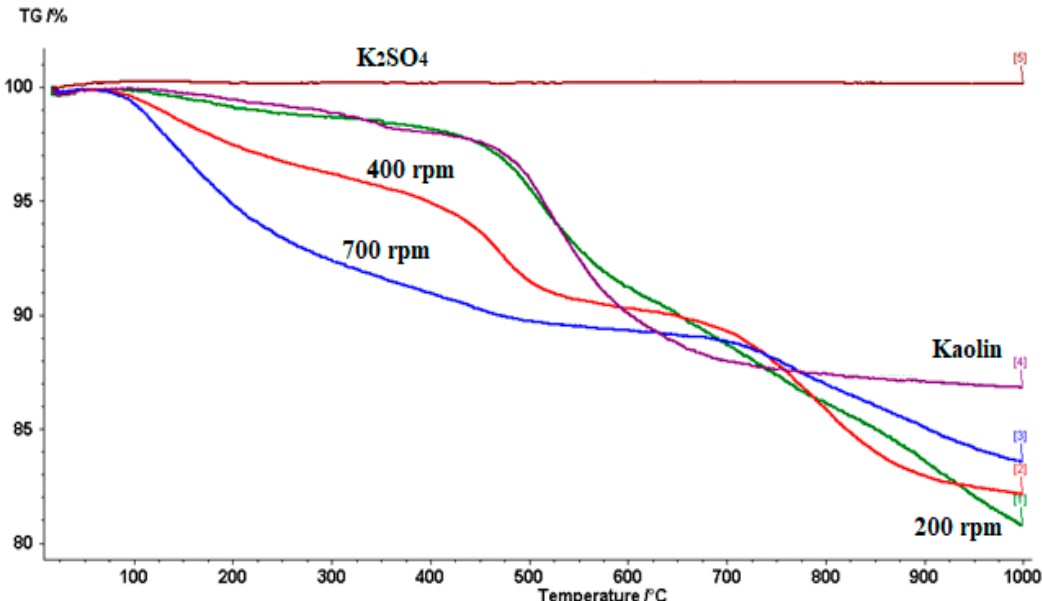

**Figure 6.** TGA patterns of kaolinite-$K_2SO_4$ samples with multiple milling speeds for 120 min.

The measurements showed that there is no mass loss in the milled starting materials samples, but there is some mass disappearance after the thermal treatment decomposition of the products. This means that new products form by the mechanochemical reaction [3].

The release of $K^+$ and $SO_4^{2-}$ ions from the kaolinite-$K_2SO_4$ mixtures milled at varied milling rotational speeds and dispersed in distilled water for 24 h is presented in Table 3 and graphed in Figure 7 for visual comparison.

**Table 3.** The concentration of $K^+$ and $SO_4^{2-}$ nutrients released from kaolinite-$K_2SO_4$ mixture milled at different rotational speeds for 120 min.

| Milling Speed (rpm) | Conc. of $K^+$ (mg $L^{-1}$) | Conc. of $SO_4^{2-}$ (mg $L^{-1}$) |
|---|---|---|
| 200 | 450.8 | 4344.1 |
| 400 | 315.2 | 3110.3 |

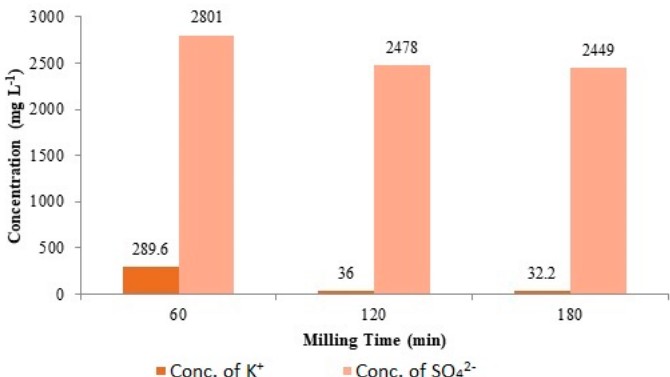

**Figure 7.** Profile of the nutrients release characteristics of milled kaolinite-$K_2SO_4$ mixtures for 120 min with different milling times, after being dispersed in distilled water for 24 h.

For sample mixtures prepared at 200 rpm, the release of $K^+$ and $SO_4^{2-}$ reached over 450 and 4344.1 mg $L^{-1}$, respectively. This relatively high value states that higher milling rotational speeds are required for a better-controlled release of the nutrients from prepared kaolinite-$K_2SO_4$ mixtures. When the milling speed was increased from 400 to 700 rpm, the release of $K^+$ and $SO_4^{2-}$ nutrients into solution significantly decreased, reaching around 315 and 3110 mg $L^{-1}$ at 400 rpm, and around 258 and 2478 mg $L^{-1}$ at 700 rpm, respectively.

The lower release values indicate that $K_2SO_4$ was integrated into the amorphous structure of kaolinite. The system of Al-Si–O, particularly with the presence of metals such as Al, retards the diffusion movement of those elements.[20] Ions of $K^+$ and $PO_4^{3-}$ in water come from both $KH_2PO_4$ and those attached in the network chains of Al-Si–O.

## 4. Conclusions

The obtained results in the present article confirm that new mixtures of kaolinite-$K_2SO_4$ can be produced by utilizing green chemistry and act as slow-release fertilizers when used in the agriculture sector. The mechanochemical process is successfully used to synthesize a novel SRF integrated with $K^+$ and $SO_4^{2-}$ ions as plant nutrients by mixing and milling of $K_2SO_4$ with kaolinite as a clay mineral. The milling time and speed parameters for the raw materials are found to be significant in theh solid-state mechanochemical preparation process. When dispersed in distilled water for 24 h, the nutrients of $K^+$ ions are released from the kaolinite-$K_2SO_4$ sample system and leached into the solution reaching around 32 mg $L^{-1}$ for preparation conditions of 180 min and 700 rpm. The foregoing condition of production refers to the optimal milling time and speed for kaolinite-$K_2SO_4$ SRF system. Additionally, it was found that temperature is an effective factor in incorporating $K_2SO_4$ into the amorphous structure of the as-received kaolinite in the production of fertilizers by the mechanochemical route. The SRFs will find increasing future applications due to the slower and controlled release of fertilizers.

**Author Contributions:** Conceptualization, E.A., M.A., I.H. and Q.A.-A.; Methodology, E.A., M.A., I.H. and Q.A.-A.; Software, M.A., Q.A.-A. and H.A.-F.; Validation, E.A., M.A., I.M., Q.A.-A., H.A.-F., I.H. and A.A.-R.; Formal analysis, E.A., M.A., I.M. and Q.A.-A.; Investigation, M.A., Q.A.-A. and H.A.-F.; Resources, E.A. and I.H.; Data curation, M.A., Q.A.-A. and H.A.-F.; Writing — original draft, M.A.; Writing — review & editing, E.A., I.M., Q.A.-A., I.H. and A.A.-R.; Visualization, E.A. and M.A.; Supervision, E.A., I.M. and I.H.; Project administration, E.A. and A.A.-R.; Funding acquisition, E.A. All authors have read and agreed to the published version of the manuscript.

**Funding:** This research was funded by THE UNIVERSITY OF JORDAN and TAFILA TECHNICAL UNIVERSITY.

**Data Availability Statement:** Not applicable.

**Conflicts of Interest:** The authors declare no conflict of interest.

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
