# Peer review of "Mechanochemical Preparation of a Novel Slow-Release Fertilizer Based on K2SO4-kaolinite"

_agronomy, doi:10.3390/agronomy12123016_

Round 1

Reviewer 1 Report

Dear Authors

Your article is devoted to Mechanochemical preparation of a novel slow-release fertilizer based on K2SO4–Kaolinite.

In my opinion the topic of manuscript is interesting and also important from an environmental point of view for the agricultural applications. However, I do not recommend publishing the manuscript in this form.

I have fundamental reservations about the analysis of the results:

Page 3. Line 99: What type of kaolin was used? Please state the chemical composition of kaolin (oxide content).

             Line 103: What material are the grinding vessels made of? If they are metallic, could the sample have been contaminated?

             Line 120: From the point of view of the experiment, is it appropriate to use only distilled water for the leaching? For comparison, it is also suitable to use, for example, rainwater.

In the entire manuscript, it is necessary to add a more professional description of the FTIR analysis for reported absorption bands in all samples.

Page 4. Line 147-153: It is necessary to add a more professional description of the XRD paterns for reported samples. Page 153 …amorphous phases - the conclusions need to be supported by literary sources.

In the entire manuscript, it is necessary to add a more professional description of the thermal analysis. Figure 3 and 6: K2SO4 does not show any change on the TG curve, this means that it does not decompose, is this correct?

Page5. Line 156…: Please don't confuse TG analysis with DSC analysis…. the Thermogravimetric system revealed that the compound K2SO4 decomposed in three steps (?) , and showed endothermic peaks at temperatures of 100, 490,  and 707 °C.

Author Response

Dear Reviewer,

Thank you very much for your invaluable comments and suggestions.

Below, please find answers and explanation to your queries.

Page 3. Line 99: What type of kaolin was used? Please state the chemical composition of kaolin (oxide content).

It is white kaolin. The kaolin used in the study is from local sources and many researchers have already analyzed it. Therefore, we have added a recent reference that almost comprehensively studied different types of local kaolin [ref. 23].

Line 103: What material are the grinding vessels made of? If they are metallic, could the sample have been contaminated?

Both the grinding vessels (containers) and balls material are made of zirconia. Therefore, there should be no contamination of the samples. According to our XRD, IR results, no sample contamination have been detected.

We have added the type of material to the text (Line 103).

Line 120: From the point of view of the experiment, is it appropriate to use only distilled water for the leaching? For comparison, it is also suitable to use, for example, rainwater.

We appreciate your suggestion to use other sources of water in the leaching experiments. Our goal was to prove the concept. We assume that using rainwater, for example, will give very similar results but needs more control on the atmospheric conditions. We have used distilled water in the study since it contains very little salts with fixed properties all the time. In a future research more types of waters must be tested. 

In the entire manuscript, it is necessary to add a more professional description of the FTIR analysis for reported absorption bands in all samples.

We agree with this point. More analysis and description was added about the FTIR analysis for absorption bands in the text.

Page 4. Line 147-153: It is necessary to add a more professional description of the XRD paterns for reported samples. Page 153 …amorphous phases - the conclusions need to be supported by literary sources.

We agree with the reviewer on this point. More analysis and description was added about the XRD patterns for reported samples. Please see the text. The conclusion of amorphous phases was supported by some new references (line 153).

In the entire manuscript, it is necessary to add a more professional description of the thermal analysis. Figure 3 and 6: K2SOdoes not show any change on the TG curve, this means that it does not decompose, is this correct?

In the studied range of temperature, according to our results and the literature, K2SO4 does not decompose (it is a salt) and it only loses water of hydration. References were added to the text to support this.

Page5. Line 156…: Please don't confuse TG analysis with DSC analysis…. the Thermogravimetric system revealed that the compound K2SO4 decomposed in three steps (?) , and showed endothermic peaks at temperatures of 100, 490,  and 707 °C.

We thank the reviewer for spotting this unintentional error, The description was corrected to state that that the mixture have decomposed and not the K2SO4. (page5, Lines 158-159).

Reviewer 2 Report

attached

Author Response

We thank the reviewer for his/her invaluable comments and below are our response to the queries and suggestions raised by the reviewer.

General comment: More details were required to describe the material and methods, in general, and in particular the leaching experiment and the type of soil used. From the part written in line 120, I understood that there was a leaching experiment in the soil. Is it correct? Please clarify. Also, make sure you used IS unit throughout the manuscript.

The description of the leaching process was corrected in the text (page 3) and we confirm that the SI units are used throughout the manuscript.

Specific comments:

Line 135: from 60 to 180 min. at….. Please remove the dot after “min” Do NOT add a dot after min in all manuscripts.

The dot was deleted after min in all wrong position in manuscript. Note that some min come at the end of sentences.

Line 135: … at a milling speed. Please change it to “..at a fixed milling speed.”

Corrected.

Line 147: The XRD patterns in Figure 2…

Added.

Line 149: Please try to find a reference that supports your explanation.

Three supporting references were added to explain the sentences [25-27].

Line 160-161: Also, the 160 figure reveals the presence .. Please change “figure” to “TAG patterns reveal the…”

Done.

Thank you again.

Round 2

Reviewer 1 Report

Dear authors, I take note of your comments and I agree, but I have one more important comment: please correct the sentences in line 159-160, it is stated that TG curves showed… endothermic peaks at temperatures…. But in the figure 3 are not  DSC curves (endothermic or exothermic peaks), only TG is shown.

Author Response

Dear Reviewer,

Thank you very much for your agreement to our modifications and for your last comment. We agree with you that they are TG curves and not DSC. Therefore, and as a result of your kind comment, we have modified the text (lines 159-160) and also we improved Figure 3 to be clearer to readers.

We hope now that you will be satisfied with all changes made.

Best regards,

Authors

Reviewer 2 Report

please see attached file.

Author Response

Again, we thank the reviewer for his/her invaluable comments and below are our responses to the queries and suggestions raised by the reviewer.

Regarding the leaching experiment, we confirm that only batch experiments (water) were conducted to study dissolution behaviors (i.e. without soil).

We agree with the reviewer that “ppm” is not an SI unit; therefore we have rechecked the units throughout the manuscript and corrected them into SI units. We used mg L-1 as suggested by the reviewer. Changes included the text, figures, and tables.

Moreover, more details were added to the “Materials and Methods” section to clarify the materials and methods, particularly about the acquisition of X-ray diffraction patterns.
